# Design and fabrication of flexible DNA polymer cocoons to encapsulate live cells

Tao Gao [1,2], Tianshu Chen[1], Chang Feng[3], Xiang He[4], Chaoli Mu[3], Jun-ichi Anzai[5] & Genxi Li[1,3]

The capability to encapsulate designated live cells into a biologically and mechanically tunable polymer layer is in high demand. Here, an approach to weave functional DNA polymer cocoons has been proposed as an encapsulation method. By developing in situ DNA-oriented polymerization (isDOP), we demonstrate a localized, programmable, and biocompatible encapsulation approach to graft DNA polymers onto live cells. Further guided by two mutually aided enzymatic reactions, the grafted DNA polymers are assembled into DNA polymer cocoons at the cell surface. Therefore, the coating of bacteria, yeast, and mammalian cells has been achieved. The capabilities of this approach may offer significant opportunities to engineer cell surfaces and enable the precise manipulation of the encapsulated cells, such as encoding, handling, and sorting, for many biomedical applications.

[1] Center for Molecular Recognition and Biosensing, School of Life Sciences, Shanghai University, 200444 Shanghai, P.R. China. [2] Jiangsu Key Laboratory for Molecular and Medical Biotechnology, College of Life Sciences, Nanjing Normal University, 210023 Nanjing, P.R. China. [3] State Key Laboratory of Pharmaceutical Biotechnology, School of Life Sciences, Nanjing University, 210023 Nanjing, P.R. China. [4] School of Biomedical Engineering, Shanghai Jiao Tong University, 200240 Shanghai, P.R. China. [5] Graduate School of Pharmaceutical Sciences, Tohoku University, Aramaki, Aoba-ku, Sendai 980-8578, Japan. Correspondence and requests for materials should be addressed to G.L. (email: genxili@nju.edu.cn)

The naturally evolved landscape of cell surfaces is assembled for cell connection, communication, and synergetic bio-functions. Encapsulation of live cells into a tunable and biocompatible surface layer can thus lead to advancements in a variety of application fields by providing cell with additional functions, offering various arrangements, mimicking the extra-cellular matrix, and controlling cell differentiation[1–4]. Over the past decade, polymer-based encapsulation methods have been developed to meet these requirements by offering cells with the anticipated functions, interactions and mechanical properties[5–7]. Hence, the engineering of cell surfaces with synthetic polymers has been a powerful strategy to expand the molecular landscapes, and is amenable to cell encapsulation that requires flexible and tunable handling for advanced applications.

Till now, many polymer-based approaches have been introduced to encapsulate cells on the basis of layer-by-layer[8], cell-in-shell[9,10], and cell-in-microgel strategies[11,12]. These strategies have introduced various functions to cells, but the modification processes are often biologically incompatible. And the polymer shells are usually stiff and thick, which may inhibit the cell capabilities[13], such as signal transduction and mass transport. Presently, efforts have been made to address the issues of cell viability and function maintenance in several polymerization approaches, such as mussel-inspired chemistry[14,15], fast kinetic gelation[12], and biomolecular assembly[16]. These approaches have been used to fabricate functional cell envelopes, but the issues of low encapsulation efficiency and uncontrolled polymerization reactions have not been fully addressed to meet the requirements. Most recently, direct and in situ encapsulation methods have been developed to improve coating efficiency and to reduce polymer thickness on the basis of the new chemical polymerization approaches[17,18]. Nevertheless, in many aspects of the polymer-based encapsulation approaches, significant challenges still remain. (1) Cell viability can be threatened by any of the toxic polymer monomers or harsh reaction conditions involved in the polymerization system; (2) uncontrolled reaction processes can result in cell aggregation, a high polymer-to-cell ratio, or low polymer grafting efficiency; (3) the grafted polymers are usually resistant to be post-tailored; and (4) more importantly, the manipulation of the surface-grafted polymers and polymer-encapsulated cells with high precision is required but is difficult at the present state-of-the-art.

Thus far, we have noted that little work has reported the use of biosynthetic reactions to fabricate biopolymer shells on cells. Furthermore, post-tailoring and post-editing of local properties of the polymer layer at cell surface has not been addressed. Therefore, in this work, we have proposed a biosynthetic approach to weave DNA polymer cocoons on live cells, by developing the in situ DNA-oriented polymerization approach, isDOP. The naturally synthesized biopolymer, deoxyribonucleic acid (DNA), is in situ synthesized as the coating material, not only for its biocompatibility and bioorthogonal polymerization process[19] but also for its tunable properties, which come from chemically synthesized nucleotide analogs[20,21], substitutive backbones[22,23], and DNA-modifying enzymes[24,25]. These properties may offer a variety of functional groups to be engineered at cell surfaces, and also extend our ability to nourish and handle these cells[26]. Furthermore, combined with the recent research progress in DNA isothermal replications[27–29] and dynamic assemblies (or reactions) of the DNA polymers, the interfacial interactions of DNA strands and assembled structures[30–34] have added functions to membranes[35], facilitating cell–cell interactions[30,36], surface bio-marker profiling[37,38], and molecular events monitoring[39,40]. In this work, by integrating DNA isothermal replication and programmed DNA assembly, the polymer density, mechanical properties, and surface chemistry can be tailored. Most importantly, the DNA polymer is a molecularly precise assembly of high homogeneity that may provide addressability with nucleotide by specific DNA base pairing (A–T and G–C) and the assistance of DNA-modifying enzymes. Therefore, isDOP presents a highly tunable technique to address cell encapsulation challenges, and it is anticipated to provide flexible mechanisms for manipulating biophysical and physiological phenomena at cell interfaces.

## Results

**Principle of isDOP.** DNA replication is a biological process of polymerization that strictly maintains the fidelity of biological inheritance. Inspired by this precise biosynthetic process, we have moved the in vivo DNA replication to cell surface, so that DNA are synthesized and precisely assembled into polymer networks, addressing the cell encapsulation challenges. In the DNA-orientated polymerization approach (Fig. 1a), two isothermal and enzymatic polymerization reactions are involved to fabricate the DNA polymer network: the rolling cycling replication (replication 1, R1) and the branched replication (replication 2, R2), which are respectively seeded by two sets of primers, the initiating primer (IP) and the branched primer (BP).

To assemble the DNA network at cell surface, we have performed in situ DNA-orientated polymerization (isDOP) (Fig. 1b). Here, the initiating primer (IP) is attached to cell membrane[41,42], so isDOP is started at the site of IP. The R1 and R2 reactions then guide the assembly of the DNA cocoons at cell surface. Specifically, IP initiates R1, which generates long and periodic DNA polymers (called the longitude DNA, LonDNA) when we introduce a single-stranded circular DNA (cirDNA) as the replication template. Then, the BP initiates R2 that generates the second kind of single-stranded DNA polymers (called the latitude DNA, LatDNA), which leads to the assembly of connections across these initial polymers (LonDNA) based on the design of replication templates. LonDNA and LatDNA are automatically cross-assembled during the replication processes, and the DNA cocoon is thus fabricated in situ at the cell surface (Fig. 1c).

**Characterizations of the R1 and R2 reactions.** Agarose gel analysis of the DNA products is used to show the feasibility of the polymerization system. In Fig. 2a, gel analysis reveals that the R1 produces extremely long LonDNA (>10 kbp, lane 1), while the independently performed R2 is unable to produce LatDNA (lane 2). However, a clear and bright band has been observed when R1 and R2 are coupled (R1R2) (lane 3), indicating that additional DNA polymers are synthesized. Atomic force microscope (AFM) observations further confirms the fabrication process, in which the structural details of the DNA polymers are revealed. R1 produces extremely long LonDNA (>3 μm, Fig. 2b), and after coupling with the branched replication, the DNA polymers spread into a fan shape network (Fig. 2c), finally forming the branched and cross-linked DNA networks (Fig. 2d).

How R1 and R2 guide the fabrication of the DNA network is further investigated by a nuclease degradation test, in which the products of R1 and R1R2 are separately degraded by the S1 nuclease, an enzyme that specifically degrades single-stranded DNA[43]. Therefore, we know whether the product is a mass of single-stranded DNA polymer or the cross-assembled double-stranded DNA networks. In the S1 degradation test, gel analysis in Fig. 2a shows R1 is sensitive to the S1 nuclease, in which the product of R1 is degraded into nucleotides, which are merely observed (lane 4). On the contrary, the product of R1R2 is strongly resistant to S1 nuclease degradation, and a bright gel band over 10 kbp has been observed (lane 7). Additionally,

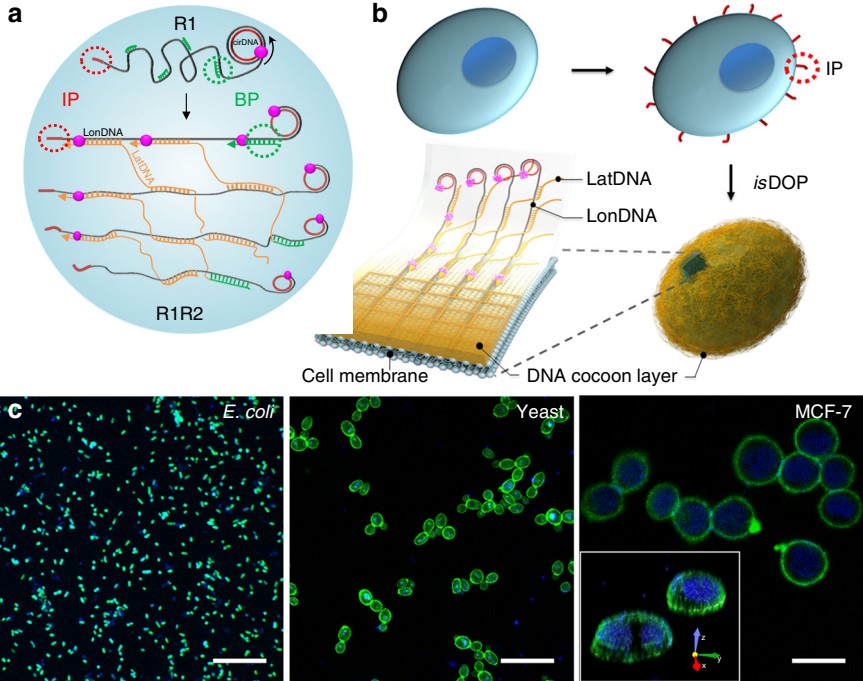

**Fig. 1** In situ DNA-oriented polymerization reaction (isDOP) for cell encapsulation. **a, b** The isDOP contains two DNA replication reactions, R1 and R2. R1 is primed by the initiating primers (IPs), leading to the assembly of long initial polymers (LonDNA, gray). R2 is primed by the branched primers (BPs), leading to the branched replication that synthesizes the LatDNA polymers (yellow). Sequence-specific assemblies across these LonDNA and LatDNA polymers fabricate the DNA cocoon at cell surface. **c** The scanning confocal microscope images show that typical cell types are encapsulated in the DNA polymer cocoons, including bacterial (*E. coli*), eukaryotic (yeast), and mammalian cells (MCF-7). The insert shows the reconstructed 3D-image of encapsulated MCF-7 cells. The surface-grafted DNA cocoons are labeled with FAM-modified oligonucleotides (green). The cell nuclei are stained with Hoechst 33342 (blue). Scale bar, 20 μm

periodic ladder tails with defined lengths are shown, indicating the repetitive nature of the DNA networks from rolling circle polymerization. Therefore, we assume that the R1 reaction generates the single-stranded DNA polymer (LonDNA), while after coupling with R2, a highly structured DNA network is fabricated. This assumption has been further confirmed by the enhanced resistance to S1 nuclease degradation when we enhance R2 by adding more branch primers (BPs) (lanes 5–7).

In addition, a more specific experiment has been developed to analyze the relationship of the two reactions on cells. As indicated in the schematic illustration in Fig. 1, two primers (IP and BP) respectively initiate the polymerization reactions (R1 and R2) to synthesize the LonDNA and LatDNA strands. So, we have respectively labeled the two strands with fluorescent dyes (FAM and TAMRA)-modified primers. Colocalization of the two fluorescent signals reveals cross-connections of the LonDNA and LatDNA strands (Fig. 2e). Therefore, it is believed that the DNA polymerase acted as a loom to weave LonDNA and LatDNA; further guided by the sequentially and mutually aided R1 and R2 reactions, LonDNA and LatDNA polymer strands are fabricated into a DNA polymer network.

**Attach the IP to the cell surface**. To enable DNA polymerization directly on the cells, the IP has been attached to the cell surface by employing the cell walls and membrane compounds as the anchor sites. Two forms of interactions, covalent ligation (for *E. coli* and yeast cells) and noncovalent insertion (for mammal cells), are used to attach the IP to the cell surfaces on the basis of the 5′-end modifications (SDA and DSPE-PEG2000)[39,42] (Supplementary Fig. 1a). The efficient anchoring of IP is observed by using a fluorescence microscope after incubating the mammalian cells (e.g. MCF-7) with a 6-carboxy-fluorescein (FAM)-labeled IP, F-IP (Supplementary Fig. 1b). The anchoring efficiency has been

revealed by flow cytometric evaluation, where the serial dilutions of the F-IP are incubated with the cells. Here, assuming that the cells have a round shape and the detected fluorescent intensity is linearly corrected with the amount of the IP, a standard calibration curve is established on the basis of cell fluorescence intensities at each concentration (Supplementary Fig. 1c and 1d).

To calculate the number of anchored IP, the cells are first incubated with F-IP. After centrifuge washing, the cells are collected and then incubate with a micrococcal nuclease that could cut off the surface-attached F-IPs, releasing free fluorophore into the solution. The amount of attached F-IP is determined according to a calibration curve of standard F-IP concentrations (Supplementary Fig. 2). Approximately $1.3 \times 10^7$ molecules are calibrated per cell when incubated with 400 nM F-IP. The surface density of the attached IP could be adjusted from $10^5$ to $10^7$ molecules per cell. The calculation method and Eq. (1) are shown in the Methods. Stability test shows these surface-anchored IPs are stable during the encapsulation process (Supplementary Fig. 3).

**Fabrication of the DNA cocoons on the cells**. IP and BP have been found to be the influential factors when fabricating DNA cocoons at cell surface, as they determine R1 and R2 reactions in isDOP. As shown in Figs. 3a, b, the DNA network is not formed at low IP density. DNA patches instead of well-aligned DNA polymer networks are formed when we incubate cells with 10 nM of IP. As a control, we solely conduct R1. In this case, small DNA polymer dots are observed (Fig. 3a), which are different from the DNA patches that are generated by the coupled reactions of R1R2 (Fig. 3b). Therefore, it is speculated that the limited number of initiation sites (IP) inhibit the formation of the DNA cocoons, possibly because the isolated LonDNA strands are too far to be bridged by the LatDNA strands at the cell surface. According to

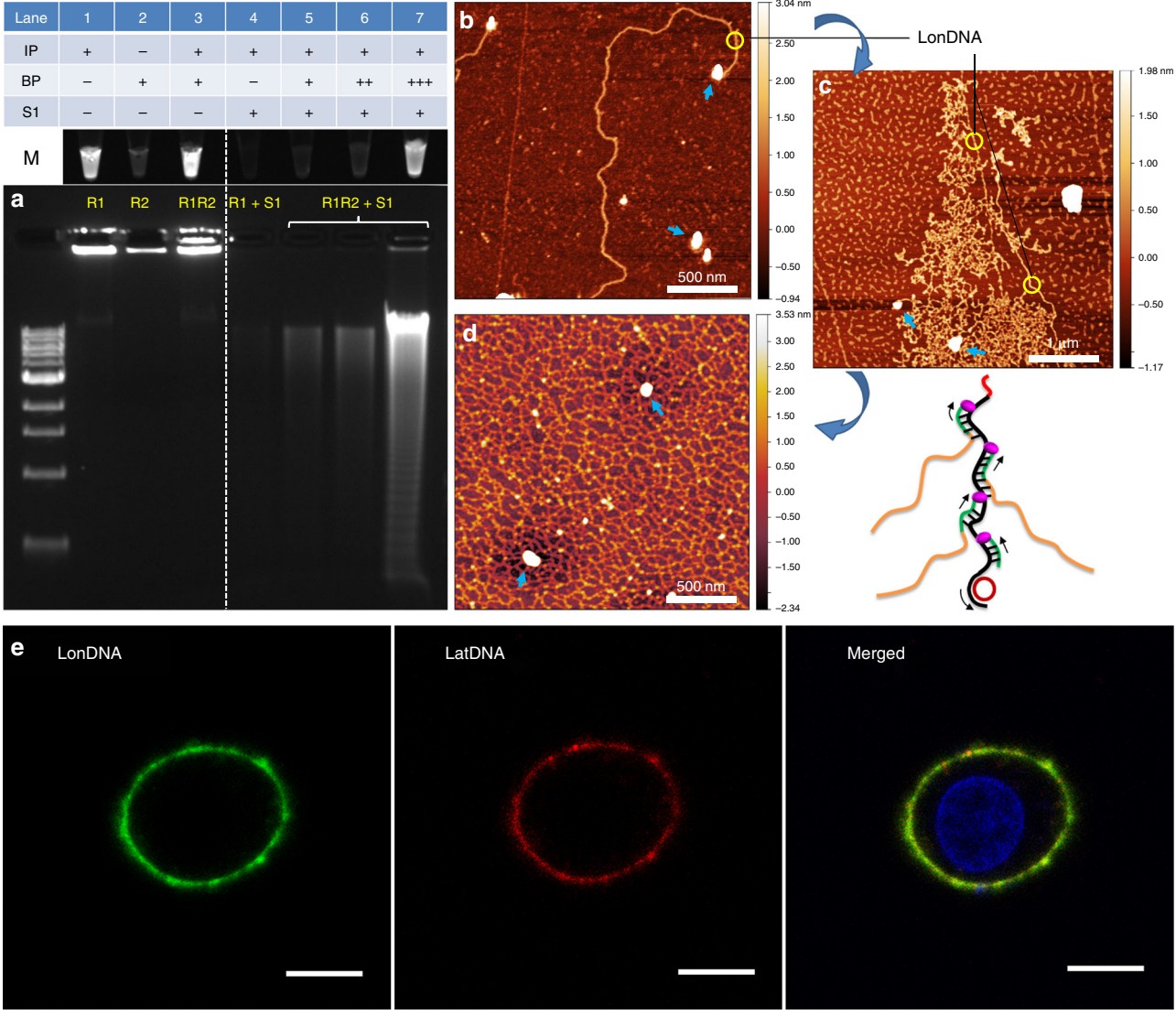

**Fig. 2** Characterizations of the R1 and R2 reactions. **a** Feasibility tests of the polymerization reactions using agarose gel analysis. The R1 and R2 reactions are performed individually or coupled. Lanes 1–3 respectively show the DNA products of R1, R2, and R1R2 after staining with GelRed. Lanes 4–7 show the S1 nuclease degradation test to reveal the feasibility of DNA assembly during the reaction. The increased concentrations of BP (5, 25, and 100 nM) in R1R2 show enhanced resistance of the DNA polymer networks against S1 nuclease degradation. DNA marker, 1 kbp ladder from 0.5 to 10 kbp. Source data are provided as a Source Data file. **b−d** Typical AFM images show the fabrication process: **b** LonDNA strands generated by R1, **c** LatDNA strands crosslinking LonDNA to form a fan-shaped DNA network at the beginning of R2, **d** DNA network fabricated by the coupled reactions of R1R2. The blue arrows indicate the positions of DNA polymerases. The z-color scales in (**b–d**) are 3.98, 3.12 and 5.87 nm, respectively. **e** Scanning confocal microscope images show the locations of LonDNA and LatDNA strands at cell surface. The two strands are respectively labeled with dye-modified IP (green) and BP (red) probes. Scale bars, 10 μm

the flow cytometry analysis of the fluorescence intensities of the grafted DNA, when the IP concentration is increased to 50 nM, the encapsulation process becomes significant vs. control group ($P < 0.01$, two-way ANOVA, Fig. 3k). Thus, we find that a proper incubation concentration of the IP is required to fabricate well-aligned DNA cocoons at the cell surface.

The influence of BP has been investigated as BP determines the branch site of the DNA cocoon. According to the fluorescent observations and the corresponding fluorescence intensity analysis of the DNA cocoons on cells, DNA polymers are observed to increase when the concentration of the BP is increased (Fig. 3d–i). The flow cytometric evaluation (Fig 3j, k) has further confirmed these observations. To reveal the structural details, we conduct isDOP on mica with identical IP densities. The AFM characterizations show that the density of the DNA network increase along with the BP concentration

(Supplementary Fig. 4). The frequency analysis reveals the height changes of the DNA polymers on mica. The density of DNA polymers are increased, respectively show 7, 13, and 20 times per micrometer, and the pore size of the polymer network decreases from $2116 \pm 2700$ to $52 \pm 178$ nm$^2$ (means ± s.d., $n = 3$). The thickness of the DNA network is also observed to increase from $3.1 \pm 0.3$ to $7.1 \pm 0.8$ nm (means ± s.d., $n = 3$). Therefore, the polymer density and pore size of the DNA cocoon can be changed by regulating the concentration of BP.

To show the structure details of the fabricated DNA cocoons at cell surface, we further perform isDOP on yeast cells, followed by conducting AFM characterizations on the yeast cell wall, a relatively rigid and flat surface relatively suitable for AFM characterization[43]. Here, DNA polymer networks have been observed to spread on the cell wall (Fig. 4). But the resolution is not satisfactory compared to that on mica (Fig. 2b, c and

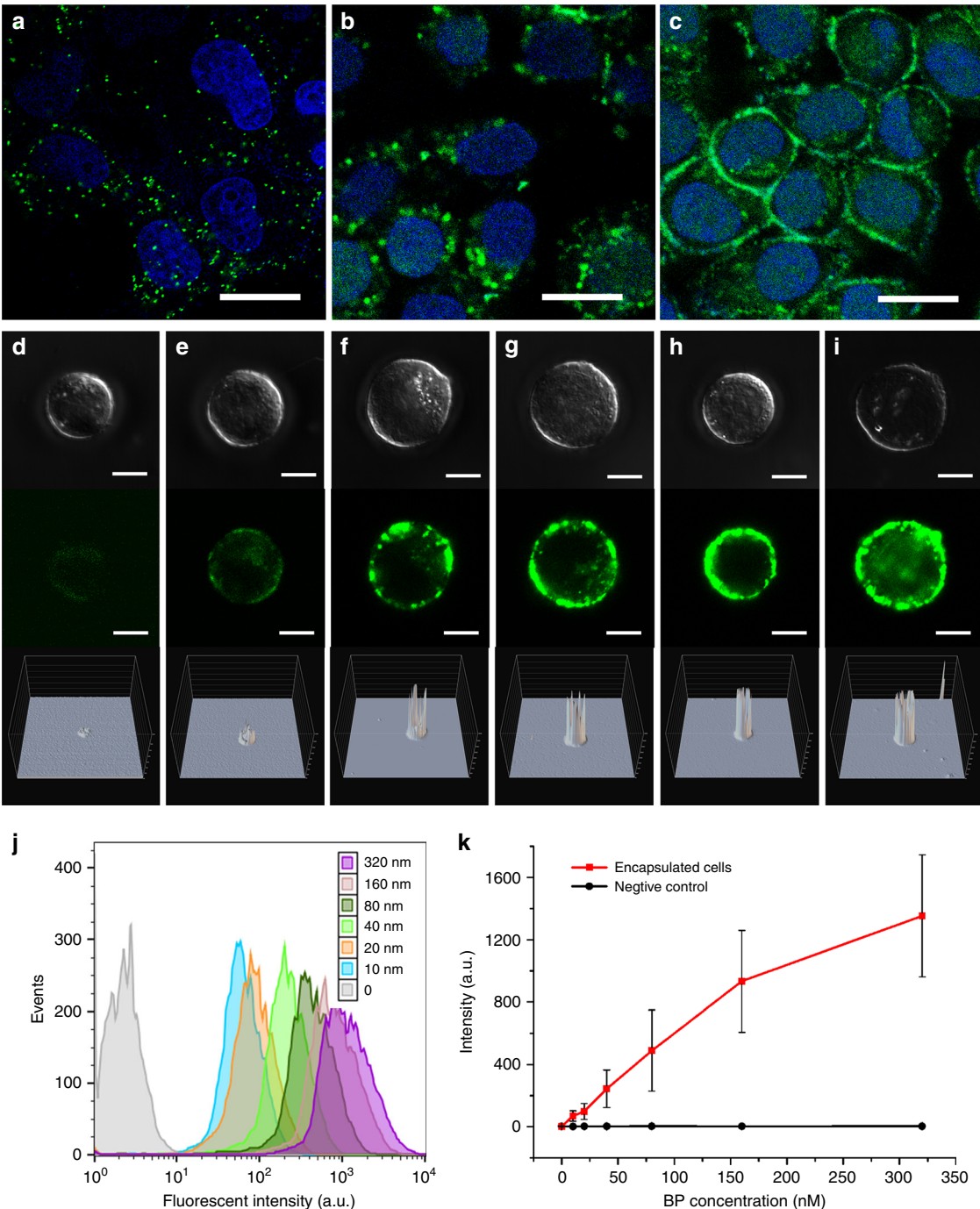

**Fig. 3** Fabrication of DNA cocoons on cells. **a**−**c** Confocal fluorescence microscopy images show the grafted DNA polymers on MCF-7 cells. The influence of the R1 and R2 reactions are investigated at low concentration of 10 nM IP, where image (**a**) shows the solely conducted R1, and image (**b**) shows the coupled R1R2 reactions. Image (**c**) shows the R1R2 reactions when the IP concentration is 150 nM. The cell-surface-grafted DNA polymers are imaged after labeling with FAM-modified oligonucleotides (green). Attached MCF-7 cells are used for the fluorescent observation in the culture dish. Scale bars, 20 μm. **d**−**i** Differential interference contrast (DIC) and confocal fluorescence microscopy images of the individual encapsulated MCF-7 cells, revealing the influence of R2 on the formation of the DNA cocoon. The concentrations of the BP in R2 are 10, 20, 40, 80, 160, and 320 nM. The bottom row shows the analysis of the fluorescent intensities, indicating the gain of DNA polymers densities in the DNA cocoon. Scale bars, 10 μm. **j**, **k** Flow cytometric evaluation of the polymer density of the DNA cocoons on the MCF-7 cells. The above BP with concentrations of 10–320 nM are used for the cell encapsulation with isDOP. Source data are provided as a Source Data file. The error bars indicate the standard deviation of 10,000 cell events at each concentration

Supplementary Fig. 4). This may be attributed to that the radius of curvature of the yeast cell (at the micrometer-scale) is three orders of magnitude larger than the height changes of DNA (at the nanometer-scale)[44]. We have further used fluorescent observations to confirm the formation of DNA network at cell surface. After encapsulation, the synthesized DNA strands have been labeled with dye-modified probes, the detected fluorescent signals have indicated cross-connecting of LonDNA and LatDNA stands to fabricate DNA cocoons (Figs. 1c, 2e, Supplementary Fig. 5, and Supplementary Movie 1).

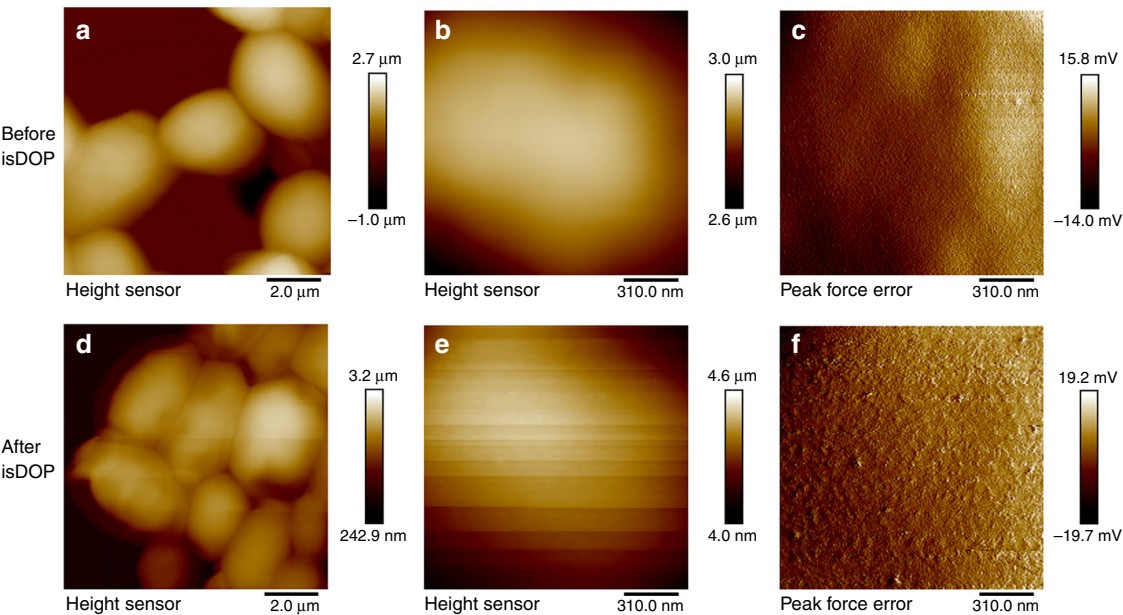

**Fig. 4** AFM imaging of the DNA cocoon on cells. **a**−**c** The typical AFM images show the surface of yeast cell walls before and, **d**−**f** after isDOP. Yeast cells are trapped in polycarbonate porous membrane for AFM imaging in the contact mode. The image scales, 10 μm × 10 μm and 2.5 μm × 2.5 μm. AFM atomic force microscope

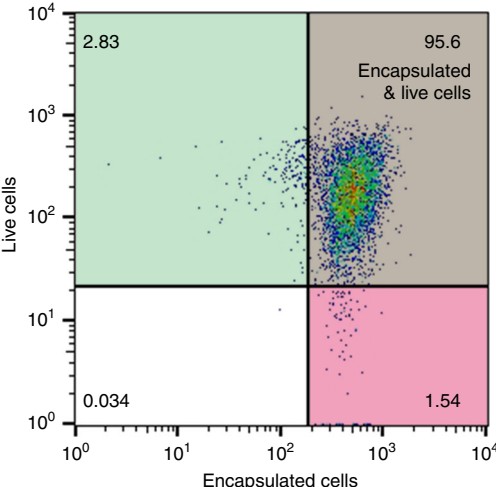

**Fig. 5** Flow cytometry analysis of the cell viability and encapsulation efficiency. The encapsulation efficiency is evaluated by staining the surface-grafted DNA polymers with PI (red). Cell viability is visualized by staining the cytoplasm with a live cell indicator, Calcein-AM (green)

**Encapsulation efficiency and cell vitality test**. Encapsulated cells with high viability are essential for the application of these cells, especially for cytoprotection, tissue engineering, and cell delivery-based therapy[17,32]. However, it has been a challenge to keep cells alive in a polymer-based encapsulation[18]. Here, in isDOP, the polymerization reactions are capable of synthesizing DNA polymers directly on the live cells in a cytocompatible buffer, where the cell culture medium is mixed to maintain cell viability. Accordingly, cell viability is minimally affected during isDOP (Supplementary Table 1, Supplementary Figs. 6 and 7). The viability assessment after encapsulation shows that over 95.6% of the MCF-7 cells are alive and encapsulated (Fig. 5), and 87.18% of the cells are single encapsulated (Supplementary Fig. 8). The results suggest that this encapsulation approach is biocompatible and highly efficient. For viability evolution over time after the encapsulation, it is found that mammal cells can lose viability if the DNA cocoon is not relieved in the long term, as a result of the anchorage-dependent nature of mammal cells and the encapsulation of DNA polymer network (Supplementary Fig. 9). For yeast and bacterial cells, fluorescent observations have shown that they are efficiently encapsulated in the fluorescent observations (Fig. 1c), with relative high viability (Supplementary Fig. 7). The medium lethal time ($LT_{50}$) is used for the assessment of cell viability and the biocompatibility of this approach. Typically, for mammalian cells (e.g. the MCF-7 cells), $LT_{50}$ is over 96 h in DMEM; the $LT_{50}$ values for encapsulated *E. coli* and yeast cells are 2 weeks or longer in the culture mediums, indicating these cells are efficiently encapsulated and kept well after encapsulation.

**Flexible encapsulation and precise handling of the cells**. Engineering the cell surface with synthetic macromolecules is a powerful approach to expand the molecular repertoire and properties of a cell. In the meantime, embedding the genetic code (A, T, G, and C nucleotides) in the DNA polymers facilitates the coding of many substrates with enormous applications[45–48]. The facile coupling of the DNA polymers or scaffold to the cell membrane can thus provide a number of strategies to deliver DNA materials to cell surfaces, making this strategy attractive for engineering cell−cell networks, developing drug-releasing bio-medical devices[33], controlling stem cell fate[49], and tissue development[50,51].

Therefore, we apply isDOP to encode the cells by fabricating sequence-specific DNA cocoons at their surfaces. As shown in the scheme of Fig. 6, this is achieved by inserting the encoding sequences (ESs) into the cirDNA templates. Therefore, templated by the cirDNA, the DNA polymer strands (LatDNA and LogDNA) are synthesized and assembled, and they automatically follow the sequence codes we expected, shown in Fig. 6a, Supplementary Table 2 and Supplementary Fig. 10. To show the feasibility, we light up the encoded cells by labeling the DNA cocoons. Different fluorescent color-coded DNA cocoons can be observed in Fig. 6b. Additionally, Fig. 6c shows the encoded cells are specifically captured in the different capture zones of a DNA-patterned slide surface, which is prefunctionalized with the

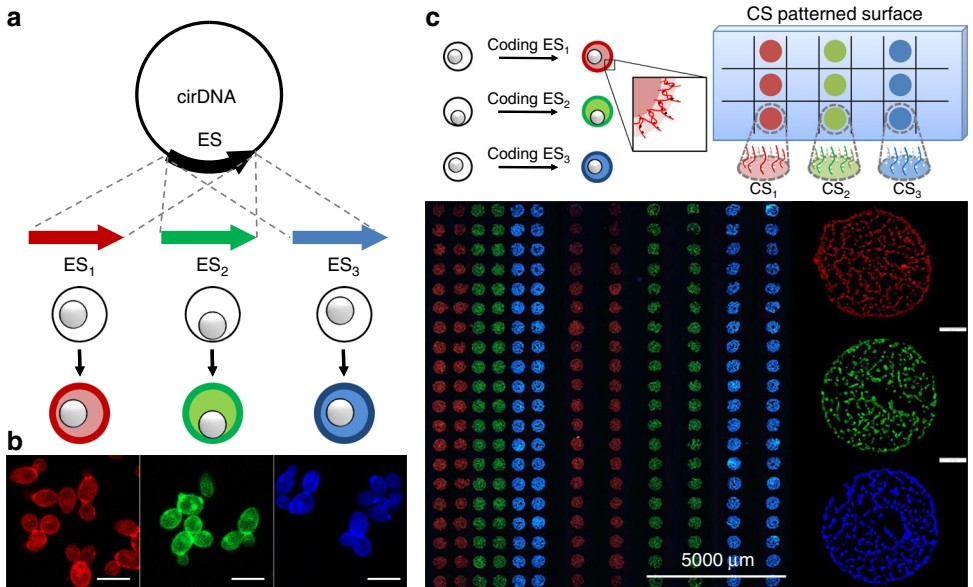

**Fig. 6** Flexible encapsulation of cells by polymer encoded DNA cocoons. **a** The encoding sequences (ESs) are inserted into the cirDNA templates. **b** The yeast cells are encoded with sequence-specific DNA cocoons. ESs (ES1, ES2 and ES3)-encoded cirDNAs are used as the replication templates. The cell-surface-fabricated DNA cocoons are labeled with the corresponding fluorescent dye (TAMRA, FAM, and AMAC)-labeled oligonucleotides, F-ESs. The sequences are shown in Supplementary Table 2, and their relationships are shown in Supplementary Fig. 10. **c** The fluorescent scanning images show that the encoded cells are captured in specific capture zones on a glass slide that is prefunctionalized with the capture strands (CS, complementary to ES). The right part shows the enlarged capture zones with captured cells, and 990 ± 86 cells have been captured at each capture zone. Data are presented as the mean ± s.d. of three independent experiments. Scale bar, 100 mm

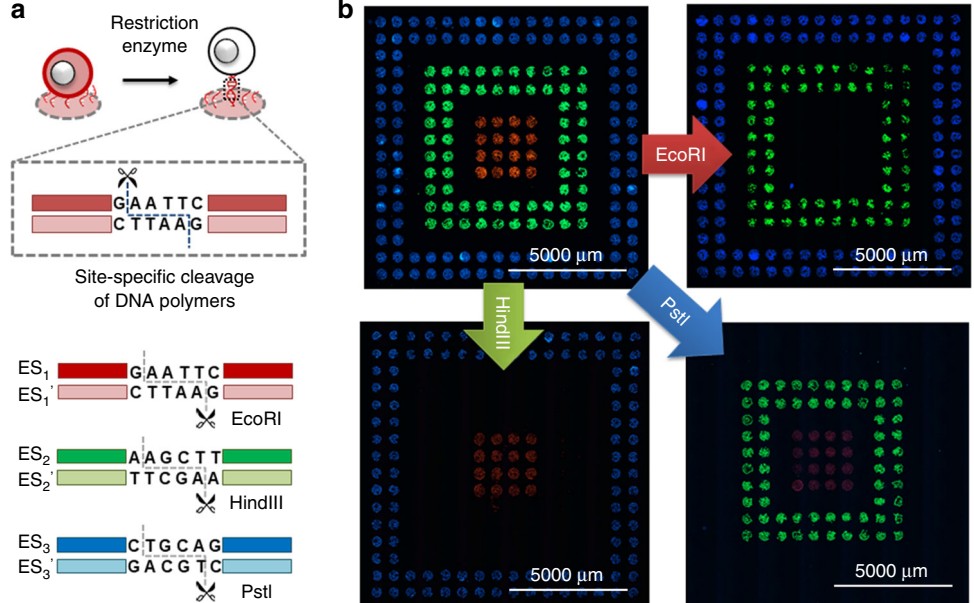

**Fig. 7** Postediting of the DNA polymer cocoons for precise handling of cells. **a** Schematic illustration showing the site-specific cleavage of the DNA polymers by the restriction endonucleases, EcoRI-HF, HindIII-HF, and PstI-HF. **b** Typical fluorescent scanning images showing the cells on the glass slide surfaces before and after the site-specific release of cells: the captured cells on the DNA sequence-patterned surface before the treatment, and remained cells on the slides after the treatments with the restriction endonucleases, EcoRI-HF, HindIII-HF, and PstI-HF respectively. Individual slides are separately used for release experiments and scanned on a cell imaging multimode reader. Cells are released from the specific capture zones as a result of the site-specific cleavage of the target DNA polymers. Scale bars, 5000 μm

corresponding capture strands (CSs). According to the calculation based on fluorescent observation, it is estimated that, on average, 990 ± 86 cells are captured at each CS site.

Post-tailoring of the surface-coated polymer is essential to handle the encapsulated cells but is often not available because many of the polymerization reactions are irreversible. Here, with

the assistance of the DNA-modifying enzymes, we can manipulate the DNA polymers after polymerization with nucleotide-level precision. As shown in Fig. 7a, Supplementary Figs. 10 and 11, the ESs are alternatively designed to serve as the cleavage sites of several high-fidelity restriction endonucleases, *Eco*RI-HF, *Hin*dIII-HF, and *Pst*I-HF. Therefore, after incubating with

**Table 1 A comparison of the different approaches for encapsulating cells in polymers**

| Encapsulation approaches | Polymer types | Encapsulation efficiency | Biocompatibility | Polymer thickness | Cell types |
|---|---|---|---|---|---|
| Layer-by-layer self-assembly[8,10,11,52,53] | Biomolecules; synthesized copolymers; extracellular matrix | High | Mostly moderate | <10 nm; 30–135 nm; accordingly | Bacteria; yeast cells; mammalian cells; tissue |
| Chemical polymerization[12,18] | Synthetic reactive monomers | High | Low | Accordingly | Mammalian cells |
| Cell-in-microgel[17,54,55] | Alginate | >90 % | High | 5.8 μm | Mammalian cells |
| Microfluidic encapsulation[17,54] | Synthetic branched polymers; biopolymers (polysaccharide, alginate, etc.) | High | High | <200 μm | Mostly Mammalian cells |
| isDOP in this work | DNA polymers | >95 % | High | <10 nm | Bacteria; yeast cells; mammalian cells |

corresponding restriction endonucleases, the targeted DNA polymers could be cleaved. The designated cells are released from the capture zones of the patterned surface (Fig. 7b). The release specificities for each of the $ES_{1-3}$ coded cells are 90.5%, 93.5% and 98.0%, respectively (Supplementary Fig. 12), indicating high releasing specificity of demanded cells. Interestingly, it is observed that the encapsulation process is reversible after releasing cells by restriction endonuclease digestion (Supplementary Fig. 13). This is because the IP would not be digested by site-specific cleavages by these restriction endonucleases. In addition, the released cells have shown relatively high viabilities during the following proliferation cultivation (Supplementary Fig. 14). Therefore, the precise manipulation of the cells is possible by the encoding and precise editing of the specific DNA cocoons with high resolution.

## Discussion

A robust approach to encapsulate cells while addressing major encapsulation challenges has been proposed in this work, on the basis of isDOP. First, a biosynthetic encapsulation strategy has been introduced for the coating of different cell types. Second, the coupled DNA polymerization/hybridization techniques have been used for cell encapsulation, so as to promote coating efficiency and accuracy. Third, by using the intrinsic properties of DNA polymer as the coating material, precise and programmed assembly of the DNA layer is achieved, making the polymer layer tunable. More importantly, cell encoding and post-manipulation have been molecularly addressed for the first time, on the basis of DNA base pairing and selectivity of DNA tool enzymes. Therefore, facilitated by isDOP, the DNA polymer networks have been grafted on typical cell types, e.g. prokaryotic (E. coli), eukaryotic (yeast) and mammalian (MCF-7) cells. These unprecedented capabilities may offer significant opportunities to engineer cell surfaces and underline the precise manipulation in many application fields. Therefore, isDOP is an unprecedented approach for the encapsulation of cells, which has not been achieved by any other approaches (Table 1).

Aside from using DNA polymers to encapsulate cells, for further development of this emerging field, much attention can be focused on approaches that use other chemically tunable natural materials for the sophisticated control of permeability, rigidity, stimuli-responsiveness of the coating layer, and finally, the ability to endow the cells with orthogonal functions, such as designed cell–cell interactions, specified assembly manners, and targeted cell delivery. We are keen to see advancements in this field that are facilitated by the polymer-based cell encapsulation approaches.

## Methods

**Chemicals and reagents**. The chemicals used for DNA modifications, sulfo-succinimidyl 1,2-distearoyl-sn-glycero-3-phosphoethanolamine-N-[amino

(polyethylene glycol)−2000] (Sulfo-PEG$_{2000}$-DSEP) was from Nanocs Lipid. Sulfosuccinimidyl 6-(4,4′-azipentanamido)hexanoate (Sulfo-LC-SDA), 0.4% trypan blue, propidium iodide (PI), calcine-AM, cell culture medium (Dulbecco's Modified Eagle Medium, DMEM, Gibco 21063), and a live/dead bacterial viability kit were purchased form Thermo Scientific. Klenow DNA polymerase and DNA modification enzymes were from New England Biolabs (NEB), including S1 nuclease, high-fidelity restriction endonucleases (EcoRI-HF, HindIII-HF, and PstI-HF), micrococcal nuclease, and corresponding buffers. Other chemicals and reagents were all analytical grade without further purification. The DNA oligonucleotides used in this work were listed in Supplementary Table 2. Unmodified DNA oligonucleotides were basically synthesized by Sangon Biotech (Shanghai) Co., Ltd.

**Preparation of circular DNA template (cirDNA)**. The cirDNAs were prepared according to the previous reported methods[55]. Briefly, by using a quick ligation kit (NEB), 10 pmol precycled DNA oligonucleotide and 50 pmol splint DNA oligonucleotide were added in 25 μL ligation buffer. The solution was heated to 90 °C for 5 min and then cooled slowly to anneal pre-cDNA and splint DNA. After bringing the two ends of pre-cDNA close by hybridization with splint DNA, the T4 enzyme mix was added. The solution was incubated at room temperature for 5 min to transform pre-cDNA into cirDNA. After inactivation, 2 μL Exo I (5 U μL$^{-1}$) and 0.5 μL Exo III (200 U μL$^{-1}$) were added, the mixture was incubated at room temperature for 1 h. The generated cirDNA was purified with the QIAquick Nucleotide Removal Kit (Qiagen), and the concentration was determined on a BioPhotometer (Eppendorf).

**Synthesis and characterization of the initiate primer (IP)**. The SDA and PEG$_{2000}$-DSPE functionalized IPs were synthesized according to our previous reported literature[41] with slight modifications. First, 2.0 nmol of 5′-NH$_2$ modified DNA oligonucleotide was incubated with 5.0 mM of Sulfo-LC-SDA (or 10 mM of Sulfo-PEG$_{2000}$-DSEP) in 800 μL of 0.5 M NaHCO$_3$/Na$_2$CO$_3$ buffer solution, pH 8.5. The mixture was shaken gently on a thermo mixer at 37 °C for 2.0 h in brown Eppendorf tubes. Unreacted reagent was washed off with a Sephadex G-25 column. Then SDA and PEG$_{2000}$-DSPE-modified DNA oligonucleotides were characterized by HPLC with an Agilent EC-18 column (2.7 μm, 4.6 × 100 mm), utilizing a linear elution gradient of 10–80% buffer B (0.1 M triethylammonium acetate, 40% acetonitrile, pH 7.0) in buffer A (0.05 M triethylammonium acetate, 5% acetonitrile, pH 7.0). IPs were collected and desalted by using a QIAquick Nucleotide Removal Kit (Qiagen, Germany). Solution containing the IP was then dried overnight using a concentrator at 4 °C. Resulting solid was dissolved in storage buffer to a final concentration of 100 μM.

**Measurement of the anchoring efficiency of IP**. A series of concentrations of the FAM-labeled IP$_1$ (F-IP$_1$, shown in Supplementary Table 2) (0, 50, 100, 150, 200, 400, 800 nM) in 500 μL 1 × PBS buffer were respectively incubated with 2 × 10$^6$ MCF-7 cells. After gentle shaking on a mixer at room temperature for 15 min, unbound IPs were removed. An additional step, the UV light activation (365 nm, 0.32 W cm$^{-2}$) was required for the attachment of IP$_2$ to E. coli and yeast cells. Cells were diluted and brought to analysis by using a fluorescent microscopy and a flow cytometry.

To determine the amount of cell-surface-anchored F-IP$_2$ on MCF-7 cells, 1 × 10$^4$ cells were incubated with 400 nM F-IP$_1$ for 30 min. Cells were collected and suspended in 100 μL buffer (0.5× PBS, containing 50 mM Tris-HCl, and 5 mM CaCl$_2$, pH 7.4). Then, ten units of micrococcal nuclease were added to the mixture. The mixture was incubated at 37 °C for 30 min, so to degrade the surface-anchored F-IP and release fluorescent dyes (FAM) into the buffer solution. Fluorescent intensities of the buffer solutions were measured on a Bio-Rad C1000 microplate reader. The concentrations of anchored IP were determined on the basis of a linear calibration curve, established by a series of dilutions of the F-IP (Supplementary Fig. 2). Therefore, the amount of cell-surface-anchored IP per cell (SA$_{IP}$) can be

calculated according to the following equation:

$$SA_{IP} = \frac{cvNA}{n}, \tag{1}$$

where $c$ is the molar concentration of IP; $v$, volume of the buffer; $NA$, Avogadro constant; and $n$, total number of the cells used for analysis.

**Fabrication DNA cocoon on cells.** Typically, the IP attached cells (MCF-7, $5 \times 10^5$ cells; yeast, $5 \times 10^5$ cells; E. coli, $1 \times 10^7$) were collected, and incubated in 100 μL of the reaction mixture: a buffer ($1 \times$ DMEM, 25 mM HEPES, 10 mM MgCl$_2$, 1 mM DTT, pH 7.2) containing Klenow DNA polymerase (1.0 units μL$^{-1}$), cirDNA (0.1 μM), and dNTPs (0.3 μM each). For R1 reaction, the mixture was held at 37 °C in the culture incubator for 30 min. For the following coupled reactions of R1R2, the reaction mixture was further mixed by pipet resuspension with a 5 μL solution containing BP, the concentration of which was changeable accordingly. The mixture was then held at 37 °C for another 60 min in the culture incubator, and then cooled to room temperature with gentle shaking. Cells were washed and suspended in DMEM buffer, and finally brought to analysis immediately. For yeast and E. coli cells, they were collected and suspended in their culture medium before analysis.

**Fluorescent microscopy imaging.** Fluorescent dyes-labeled ES oligonucleotides (F-ES$_{1-3}$) were used as the reporter probes to label DNA cocoons, on the basis of specific hybridizations of DNA oligonucleotides. The concentration of each F-ES$_{1-3}$ in the incubation buffer ($1 \times$ DMEM) was 100 nM, and was incubated with cells for 15 min in the thermal incubator with gentle shaking. The labeled cells were washed several times with $1 \times$ PBS, and brought to image on a C2 plus confocal fluorescent microscopy, Nikon. For the image of MCF-7 cell, cells were either labeled at the surface of a culture dish or in the incubation buffer according to the assay. The 3D fluorescent intensities of the DNA cocoons on cells were measured and provided by the software of C2 plus microscopy. For reconstruction 3D-images of different cell types, a serial of confocal slice images were scanned at the z-axis. These acquired images were stacked together for construction of a 3D view in the NIS software.

**AFM imaging.** For sample preparation of DNA polymers, silicon wafer (1 cm × 1 cm) was firstly fixed on the glass slide with a double-sided tape, and the upper defective layers of the silicon wafer were removed to obtain a smooth surface. Then, 300 μL of 5% APTES in methanol was incubated on a silicon wafer for 30 min for the salinization of the surface. The silicon wafer was washed and dried in an oven at 115 °C for 1 h. The polymerization DNA products of R1 and R2 reactions were respectively placed on the surface for 2 min, then washed three times with 200 μL sterilized water, finally dried with nitrogen slowly.

For sample preparation of yeast cells, polycarbonate membrane was chosen to fix the yeast cells. One milliliter of the $1 \times 10^5$ yeast solution was drawn by a syringe and filtered on a filter that has been placed with a polycarbonate membrane, glossy side up. Then they were washed three times with 5 mL sodium acetate buffer. After filtration, the polycarbonate film was taken out and naturally dried. Then it was cut into squares of 1 cm × 1 cm and fixed on a glass slide with double-sided tape for the AFM measurements.

AFM measurements were performed in air using a Bruker ICON Atomic Force Microscope and oxide-sharpened micro-fabricated Si$_3$N$_4$ cantilevers, modulus of elasticity, 0.5 N m$^{-1}$. Intelligent scan mode was selected for the measurements. The sample was roughly scanned with the parameters: scan rate, 1.0 Hz; peak force amplitude, 150 nm; scan size 10 μm × 10 μm; sample per line, 128. For high-resolution view, sample per line was increased to 1024. For the scanning of yeast cell surface, the parameters were: scan rate, 1.0 Hz; peak force amplitude, 300 nm; scan size, 10 μm × 10 μm; sample per line, 256. High-resolution observation parameters: scan rate, 1.0 Hz; peak force amplitude, 150 nm; scan size, 2 μm × 2 μm; sample per line, 512.

**Flow cytometry analysis.** To evaluate encapsulation efficiency, surface-grafted DNA cocoon was stained with PI, a DNA binding dye that is not membrane penetrable. Cell viability was indicated by a membrane-permeable fluorescent indicator (calcein-AM), which could be activated by active lipase in living cells[5]. So, after staining cells with 1.0 μM PI and 5.0 μM calcein-AM for 15 min in the cell incubator, the encapsulation efficiency and cell viability were evaluated. The stained cells were washed, collected, and adjusted to the concentration from $2 \times 10^5$ to $1 \times 10^6$, then brought to analysis on an FACSARIA II flow cytometer (BD). The light scatter channel was set on linear gains, and the fluorescence channel was set on a logarithmic scale. 10,000 cells were analyzed in each condition. All samples were protected from light and performed with three replicates.

**Cell viability test.** After encapsulation, cells were collected, suspended in their culture mediums, and distributed into 96-well plates, and kept in culture incubators before viability measurements. At each time intervals, cells in the wells were treated with 0.1 unit mL$^{-1}$ micrococcal nuclease to release cells from the encapsulated DNA cocoons, and then were brought to viability tests. The viabilities of MCF-7 and yeast cells were indicated by 0.04% trypan blue and 1% Loeffler's methylene blue, respectively. The analysis was carried on an automated cell counter (Countess, Invitrogen). The viability

of E. coli was tested with a live/dead bacterial viability kit on a fluorescence spectro-photometer (F-7000, Hitachi). Source data are provided as a Source Data file.

**Cell encoding and manipulation.** Cells were encoded by fabrication sequence-specific DNA cocoons at the surface. Specifically, 0.1 μM each of the encoded cirDNAs (cirDNA$_1$, cirDNA$_2$, and cirDNA$_3$) was added to each reaction mixture during isDOP. Therefore, cells were encapsulated with sequence-specific DNA cocoons.

In the meantime, capture strands (CS) patterned surface was prepared to capture the encoded cells. The patterns were designed on an M2-Automation microarray spotting system, and translated into patterns that were made of 400–500 μm-diameter droplets of 20 μM 5′-amine-modified CS in a spotting solution ($1 \times$ PBS, 0.005% Tween 20, pH 8.5), onto epoxysilane-coated slides (Nexterion® Slide E, Schott). The slides were then transferred into a fresh tube of 50 mL, incubated with blocking solution ($1 \times$ PBS, 1.0 mM glycine, 0.005% Tween 20, pH 8.5) at room temperature for 30 min. The obtained slides were washed twice with 0.1% SDS, three times with deionized water, and stored in a desiccator until use.

Cell manipulation, including sequence-specific capture and release, and further cultivation, were mainly based on three elements: ESs encoded DNA cocoons, CSs patterned slide surface, and DNA tool enzymes. As shown in Supplementary Fig. 11, the patterned slide was laid in a glass dish with the pattern surface on top, then it was immersed in DMEM containing these encoded cells (1:1:1 v/v/v), each concentration of the cells was $3.3 \times 10^6$ cells mL$^{-1}$. Put the dish in a cell incubator at 37 °C for 30 min, shaking it gently several times during the incubation. After cooled to room temperature, the slide was washed three times with DMEM solution containing 0.005% Tween 20 to remove unbound cells. Finally, the slide was dried on a slide centrifuge and brought to fluorescent scanning on a BioTek Cytation™ three-cell imaging multimode reader. Otherwise, the slides were respectively incubated with restriction endonucleases solutions (0.2 units μL$^{-1}$) at 37 °C for 10 min, to release the encoded cells from specific zones of the pattern surface. The released cells were diluted with culture medium for proliferation and viability tests, respectively according to the types of restriction endonucleases.

**Statistical analysis.** Data are presented as the means ± standard deviation of the mean (s.d.). Technical as well as biological triplicates of each experiment were performed. Comparison between two groups was performed by Student's $t$ test. Multiple group comparisons were determined using two-way ANOVA. A $P$ value of 0.05 was considered statistically significant. Pearson correlation coefficient ($r$ value) was calculated assuming a linear relationship between variables. The GraphPad Prism 6 and OriginPro 9.1 were used.

**Reporting summary.** Further information on research design is available in the Nature Research Reporting Summary linked to this article.

## Data availability

All data generated or analyzed during this study are included in this article (and its Supplementary Information files). The source data underlying Figs. 2a, 3k and Supplementary Figs. 1d, 2, 3, 5, 7a−c, 9, 12e and 14 are provided as a Source Data file. All data are available from the corresponding author upon reasonable request.

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

## Acknowledgements

This work was supported by the National Postdoctoral Program for Innovative Talents (Grant No. BX201600098), the China Postdoctoral Science Foundation funded project (Grant No. 2017M611532), and the National Natural Science Foundation of China (Grant Nos. 81772593, 21235003).

## Author contributions

T.G. and G.L. developed the study concept and designed the experiments; T.G. and C.F. designed the DNA sequences; T.C., C.F. and C.M. synthesized the DNA probes, and performed the experiments; X.H. conducted the cell encoding and manipulation experiments on the DNA-patterned slide; T.G., T.C., C.F., and C.M. acquired, analyzed and interpreted the data; T.G. and T.C. drafted the manuscript in close collaboration with the other coauthors; G.L. and J.-i.A. improved the manuscript.

## Additional information

**Competing interests:** The authors declare no competing interests.

