## [Peer Review File · Nature Communications]

Reviewers' comments:

Reviewer #1 (Remarks to the Author):

The manuscript describes a novel and interesting DNA polymerization approach for encapsulating live cells for applications including cell labeling, capture, and release. The manuscript is well written and is potentially of interest to a broad range of biological research fields, if the following questions are addressed.

1. The cell encapsulation time v.s. Cell viability data (Fig. S5) is difficult to interpret because the detailed protocol is not provided in the Method. As shown in Fig. S5, it seems for each cell type the viability test was performed at different time points after DNA encapsulation. One important factor that could affect the outcome is how were the cells kept before the measurements. For example, were the MCF-7 cells kept in suspension until there measured? If so, is the decreased viability due to the anchorage-dependent nature of the MCF-7 cells or the encapsulation? Beside the experimental details, this data also need viability test results from control cells (i.e., cells without encapsulation) to compare to.
2. Similarly, Fig. S6 also needs to show growth curve of control cells.
3. For the selective release of captured cells, quantitative measurements of specificity and efficiency are needed but not provided. In addition, the related Fig. 7 (b) images are also confusing- it is not clear whether the 4 images of Fig. 7(b) are representative images or the actual images obtained before and after a release experiment.

Reviewer #2 (Remarks to the Author):

The work finished by Gao et al described a DNA polymerization/hybridization-based strategy to cover three kinds of cell types including bacteria, yeast and mammalian cells with a cocoon-like DNA shell (DNA cocoons). They characterized the formation of the DNA cocoons on cell surface and proved that different encoding sequences in DNA cocoons can help cells to be selectively captured in specific capture zones on a glass slide through DNA hybridization and the captured cells can be further released after restriction endonuclease treatment. The authors claimed that this programmable and biocompatible strategy may offer significant opportunities in cell surface engineering and precise manipulation of encapsulated cells. However, I don't see many inspiring ideas or exciting discoveries in this paper. The basic techniques used here are well-studied rolling circular amplification and DNA hybridization and the authors didn't show me a good reason to make such a design, I mean, to encapsulate cell into a DNA cocoon.

Some minor problems:

Once the primers are anchored on the cell surface, are they stable enough during the encapsulation process?

If the encapsulation process is reversible once the cocoon was digested by the restriction endonucleases?

Page 7, line 164, "The concentrations of the IP in the incubation buffer were 10, 10, and 150 nM." The concentration of IP in Figure b is the same with that in Figure a?

Figure 3, the background signals are obvious different in Figure 3a, b and c. Besides, please use the same bar length.

Figure 1c, it seems that some green signals come from the cell plasma and nucleus region. How to explain this? Were some DNA strands internalized into the cells?

In Figure 1c (i), I can see many bacteria (cell nuclei were stained by blue color) that are not surrounded by green color. Are these bacteria not encapsulated? If so, it would be misleading to claim the encapsulation efficiency is 95% in Table 1, because perhaps only the encapsulation efficiency of mammalian cells is over 95% and the efficiency of bacteria and yeasts are not discussed in the current manuscript. Additionally, I feel a little confused about the "Biocompatibility" in Table 1. The author claimed that the DNA cocoon is biocompatible, but mammalian cells used in this work lost viability quickly after encapsulation. So what is the meaning of biocompatibility here? And how to quantify it?

Overall, I cannot recommend this paper to be published in Nature Communications.

Reviewer #3 (Remarks to the Author):

In this manuscript, the authors developed the in situ DNA-oriented polymerization approach to fabricate DNA polymer networks onto three kinds of cells. And they applied this method to specifically capture and release encoded cells in a DNA patterned slide surface. This work is novel and publishable, but the manuscript could be improved.

1. The schematic illustrations of the principles of isDOP showed that the LatDNAs connect different LonDNAs together. However, the AFM images in Fig.2 did not prove the connection between LonDNAs. And the agarose gel analysis actually confirmed the repetitive double-stranded DNA structure, but it did not show the crosslink between different LonDNAs. Please give more supports and explanations about the network formation.

2. From the fluorescent scanning images, the DNA-embed cells formed both aggregates and single one, and the cell aggregation could affect the precise manipulation. Please give the results about the single cell encapsulation ratio.

3. As mentioned in cell vitality test part, the vitality of DNA cocoon embed mammalian cells could not keep well. In this case, the biocompatibility of this approach for mammalian cells needs to be reconsidered.

4. Some previous works about DNA materials for cell encapsulation are not mentioned in this manuscript.

Here are some examples.

1. Nature Methods 2015, 12, 975-981.

2. Advanced Materials 2013, 25, 4714-4717

Detailed changes of the manuscript and the point-by-point response to the reviewers' comments

We would like to thank the reviewers and editors for the time, effort and insight in examining our manuscript with an extremely rewarding peer review process. We agree with the issues and concerns raised, and have conducted several additional experiments and analysis, supplemented more data, and addressed our critical points of this work in the revised manuscript. Substantial improvements over our previous submission have been made. The manuscript, we feel, is a more rigorous and insightful one.

Overall, we have made the following core improvements to the experiments and analysis:

- 1) Developing more specified experiments to investigate the fabrication details of this approach.
- 2) Construction 3D visualization of the encapsulated cells, for more clearly observation of DNA cocoons fabricated at cell surface, together with the improved quality of data in the main text and also in the Supplementary Information.
- 3) Quantitative measurement of cell release specificity by cytometry evaluation, as well as cell viability evolution over time; and assay conditions have been improved for cell viability.
- 4) Confirmation of the validity of several core concerns pointed out by the reviewers; as well as collecting more data in the Supplementary Results to support our conclusions.

Our new findings and responses to the comments are outlined in the following point-by-point response.

Reviewers' comments:

Reviewer #1 (Remarks to the Author):

The manuscript describes a novel and interesting DNA polymerization approach for encapsulating live cells for applications including cell labeling, capture, and release. The manuscript is well written and is potentially of interest to a broad range of biological research fields, if the following questions are addressed.

Many thanks for the positive comments.

1. The cell encapsulation time v.s. Cell viability data (Fig. S5) is difficult to interpret because the detailed protocol is not provided in the Method. As shown in Fig. S5, it seems for each cell type the viability test was performed at different time points after DNA encapsulation. One important factor that could affect the outcome is how were the cells kept before the measurements. For example, were the MCF-7 cells kept in suspension until there measured? If so, is the decreased viability due to the anchorage-dependent nature of the MCF-7 cells or the encapsulation? Beside the experimental details, this data also need viability test results from control cells (i.e., cells without encapsulation) to compare to.

Many thanks for the helpful comments and the question. So the detailed protocol has been provided in the Methods section, and control cells have been used in the experiments for clearer interpretation of these data (Supplementary Figs 7 and 12). In addition, since the time points for cell viability tests are different among cell types, viability results for these cell types has been separately presented, and more data have been added based on control cells and the results of additional experiments.

We also agree with the reviewer that the condition to keep cell is an important factor to determine cell viability. In our experiment, cells were kept in cell culture medium before the measurements. As indicated by the result, the encapsulated mammal cells were not kept well. So, as suggested, we have conducted additional experiments by using different cell controls to investigate the potential influential factors, including the anchorage-dependence nature of cell and the encapsulation. From the experiments, we found the reaction condition was the major cause of unexpected viability loss of mammal cells, while the anchorage-dependent nature of MCF-7 cells and the encapsulation had limited influence.

Results:

After encapsulation, flow cytometer evaluation showed high viability (95.6%) of the encapsulated MCF-7 cells. However, the encapsulated mammal cell loss viability in a relative short time after the encapsulation, typically with a medium lethal time (LT₅₀) of 11 hours. Cells were not kept very well after encapsulation. So, experiments were conducted to test if the decreased viability was due to the anchorage-dependent nature

of MCF-7 cells or the encapsulation. In the experiment, surface-attached and suspended cells were respectively used for the encapsulation, and the cells without encapsulation were used as the controls. As shown in (a) below, we found that LT_{50} for encapsulated cells (both for anchored and suspended cells) were less than 15 hours, the result indicated cell viability loss could be caused by the encapsulation. We were confusing, because it was contrary to the previous reports that DNA would not significantly affect cell viability (Nat. Nanotechnol. 2017, 12, 1183-1189; Nat. Methods 2015, 12, 975-981; Adv. Mater. 2013, 25, 4714-4717). We believed there're other influential factors. Here, it was surprisingly found that pH value of the reaction buffer was reduced after the encapsulation process. This was observed when we used the reaction buffer containing a pH indicator (phenol red). Previously in the experiments, the pH change of the buffer was not observed as the pH indicator was not included in DMEM. In this case, we optimized the reaction conditions. 25 mM HEPES was used as a buffer agent to maintain physiological pH. In addition, we had moved the encapsulation reactions to the culture incubator to provide a more favorable environment for mammal cells. By changing these conditions, we observed the LT_{50} for the encapsulated mammal cells had been pushed to over 96 hours (b), indicating the viability of the encapsulated cell was kept relatively well in the improved assay conditions.

Therefore, in the updated Results, controls have been added to Supplementary Fig. 7, the assay conditions have been specified in the Methods and Supplementary Table 2, and the corresponding discussions have been added to the text. We thank the reviewer for the help.

2. Similarly, Fig. S6 also needs to show growth curve of control cells.

Many thanks again for the helpful suggestion. So we have added control cells in the Supplementary Results. The updated figure has now been provided as Supplementary Fig. 12.

Results:

At each time intervals (0, 1, 2, 4, 6, 8, 10, 12, 18, 24 h), proliferation of the cells were monitored on a cell counter. We started with the control by seeding 3×10^5 cells in the culture medium.

Updated Fig. S12. Cell cultivation experiment after specific release of the encapsulated cells (ES₁, ES₂, and ES₃). Yeast cells with and without DNA encapsulations have been used as controls, showing as the dashed and solid black curves, respectively.

3. For the selective release of captured cells, quantitative measurements of specificity and efficiency are needed but not provided. In addition, the related Fig. 7 (b) images are also confusing- it is not clear whether the 4 images of Fig. 7(b) are representative images or the actual images obtained before and after a release experiment.

3-1. Many thanks for the helpful comment. We agree with the reviewer that specificity and efficiency of cell release are quite important for precise manipulation of these encapsulated cells, so the experiments for the specificity and efficiency have been conducted to provide these data. For efficient release, the concentration of restriction endonuclease has been optimized, it was found that 0.2 units μL^{-1} enzyme was enough to detach all cell from the anchored surface. Bright field and fluorescent images have shown all the targeted cells are released after the treatments, shown as below. The result has also been included in Supplementary Fig. 10.

Next, flow cytometry evaluation has been used to estimate the release specificity. In the experiment, each of the F-ES₁₋₃ labeled cells are recorded by corresponding laser channels after the treatment by specific restriction endonucleases. Results in Supplementary Fig. 10a show that the restriction endonuclease-assisted cell release

are highly specificity. All targeted cells can be released by restriction endonucleases, and the release specificity for each of the ES₁₋₃-encoded cells are 90.5%, 93.5%, and 98.0%. The data have been provided as Supplementary Fig. 10 in the Supplementary Information. This important point regarding release specificity has been discussed in the Discussion of the revised manuscript.

Additional results in Supplementary Fig. 10. Release efficiency and specificity of the surface captured yeast cells. **a**) Bright (i) and fluorescent (ii) observations of the slides after treatments with each of the EcoRI-HF, HindIII-HF, and PstI-HF. **b-d**) The released ES₁₋₃-encoded cells in each treatment are recorded by flow cytometry analysis. **e**) Histogram shows the release specificity for each of the ES₁₋₃-encoded cells.

Methods and Results:

To ensure the cells were efficiently released, the concentration of restriction enzymes

had been optimized. We found 0.2 units μL^{-1} enzyme was enough to release all targeting cells from the capture zones. For cytometry evaluation, enzyme activity was terminated with a stop solution (NEB #B7024) after the release treatment, the labeled F-ES₁₋₃ probes were used as fluorescent indicators for cell counting in the flow cytometry analysis. Five thousand events (ungated) were acquired per sample in acquisition-to-analysis mode. After each individual releasing treatments, the detected events for target cells were 90.5% (EcoRI-HF targeting the ES₁ encoded cells), 93.5% (HindIII-HF targeting the ES₂ encoded cells) and 98.0% (PstI-HF targeting the ES₃ encoded cells) respectively, indicating high releasing specificities for the three encoded cells.

3-2. We are sorry for the unclear presentation of images in Fig. 7b and the vague descriptions in the figure legend. Fig. 7b shows the actual images obtained before and after a release experiment. In the experiment, the image (i) shows the captured cells before releasing, where all cells are attached on the DNA sequence-patterned zones at glass slide surface. This is similar to that of in Fig. 6c. The images (ii), (iii), and (iv) show the slides after individual treatments by corresponding restriction endonucleases. To be noted, the release experiments have been separately conducted on three individual slides (i). For fluorescent scanning on a cell imaging multi-mode reader, slide centrifuge drying was performed before the scanning, which could cause cell detachment from the slide surface. So, for better presentation of these data, the legend of Fig. 7 has been provided with more details.

Reviewer #2 (Remarks to the Author):

The work finished by Gao et al described a DNA polymerization/hybridization-based strategy to cover three kinds of cell types including bacteria, yeast and mammalian cells with a cocoon-like DNA shell (DNA cocoons). They characterized the formation of the DNA cocoons on cell surface and proved that different encoding sequences in DNA cocoons can help cells to be selectively captured in specific capture zones on a glass slide through DNA hybridization and the captured cells can be further released after restriction endonuclease treatment. The authors claimed that this programmable and biocompatible strategy may offer significant opportunities in cell surface engineering and precise manipulation of encapsulated cells. However, I don't see many inspiring ideas or exciting discoveries in this paper. The basic techniques used here are well-studied rolling circular amplification and DNA hybridization and the authors didn't show me a good reason to make such a design, I mean, to encapsulate cell into a DNA cocoon.

Many thanks for the comments, and we are sorry for not expressing our idea or concept clearly. Polymer-based coating or encapsulation of cells is the upstream and bottleneck technique in many fields of biological research. Encapsulation cell into

polymers could offer microscale control in assembly of complex tissue mimics and cell matrix supports as three-dimensional environments for tissue regeneration, and also for far-reaching implications in many fields such as cell therapeutic, medical implants, programming in vivo delivery for niche modelling (Stevens et al. *Science* **2005**, *310*, 1135-1138; Mao et al. *Nat. Mater.* **2017**, *16*, 236-243). Using DNA polymers and its derivatives as the coating material can bring new opportunities to satisfy these research demands, especially for precise microscale control of cell assembly based on DNA polymerization/hybridization, which represents a significant advance as compared to the field of DNA-mediated nanoparticles self-assembly. Here, the DNA polymer and its manipulation techniques have shown extraordinary advantages to extend our ability. For example, the spatial and hierarchical assembly of cells for controlled cell communication can be used to investigate the stem cell differentiation under the desired interactions with the supporting matrix and cells that are arranged with orders. Such kind of research is critically needed in regenerative medicine and the tissue developing on a chip. On the other hand, the key technical bottleneck leading to these implications is how to efficiently encapsulate and precisely manipulate demanded cells for downstream use. Basically, in our present work, encapsulation cells with the precisely controlled and encoded DNA polymer cocoons, can be quite helpful to promote these researches.

In order that this reviewer may clearly know the main creativity and finding of this work, please allow us to highlight this manuscript. In this work, we have developed a versatile strategy to address key encapsulation challenges, including uncontrolled polymerization process, threatens to cell viability, and lacking abilities to post-control of the coating polymers. By introducing a biosynthetic approach vs. chemical or physical approaches, the biosynthesized DNA polymers act as the coating material for cell encapsulation, which enables precise manipulation of the polymer layer both before and after the encapsulation process. Significant points of this work can be illustrated as follows:

- 1) A biosynthetic encapsulation strategy has been introduced for the coating of different cell types.
- 2) The coupled DNA polymerization/hybridization techniques have been used for cell encapsulation, so as to promote coating efficiency and accuracy.
- 3) By using the intrinsic properties of DNA polymer as the coating material, precise and programmed assembly of the DNA layer is achieved, making the polymer layer tunable.
- 4) Cell encoding and post-manipulation have been molecularly addressed for the first time, on the basis of DNA base pairing and selectivity of DNA tool enzymes.

Based on the comments of this reviewer, we have improved corresponding presentations in the Conclusion of the revised manuscript.

Some minor problems:

Once the primers are anchored on the cell surface, are they stable enough during

the encapsulation process?

Stability of the primers is indeed important because the DNA polymerization reactions are initiated from these surface anchored primers. So, in addition to the anchoring efficiency test in Supplementary Fig. 1, we have conducted stability test of the primer, the data have now been provided in Supplementary Fig. 3, which indicates the primers are stable during the encapsulation process. The corresponding discussion has been added to the text.

Results:

Anchoring efficiency of DNA primers have been testified in Supplementary Fig. 1. We further tested if the anchored primers were stable enough that they would not drop off during the encapsulation process. In the experiment, a fluorescent dye-labeled primer was used as the probe, 5' - DSPE-PEG₂₀₀₀ - TTTTT(FAM)TTTTTAGACTATATGACA - 3'. After encapsulation, fluorescent intensities of the collected cells were monitored to track if the anchored primers were drop off. We found the signal change of cells was not significant before and after the encapsulation, indicating the anchored primers were stably linked to cell and they would not drop off during the encapsulation process.

If the encapsulation process is reversible once the cocoon was digested by the restriction endonucleases?

Many thanks again for the helpful question. So we have tested the possibility of reversible encapsulation after cell releasing. The surface-anchored primers (IP) were not be digested since the sequence of IP are none of the substrates of the three restriction endonucleases. The attached IP can initiate another round of DNA polymerization reactions to reversibly encapsulate cells. In the experimental, we have observed that the encapsulation process is reversible. This extended ability has been discussed in the revised manuscript, and the result has been added as Supplementary Fig. 11 in the Supplementary Result. We sincerely thank the reviewer for the helpful question.

Methods and Results:

In the experiments, the DNA cocoon was digested by 0.5 units μL^{-1} HindIII enzyme for 15 min. Fluorescent signal showing the DNA polymer was disappeared as a consequence of the digestion. After washing, MCF-7 cells were brought for another round of DNA polymerization reactions. After labeling with F-ES₂, it was observed the cells could be coated with DNA cocoon, indicating the re-encapsulation of the cell.

Page 7, line 164, "The concentrations of the IP in the incubation buffer were 10, 10, and 150 nM." The concentration of IP in Figure b is the same with that in Figure a?

Yes, the concentration of IP in Figure 3b is indeed the same as that in Figure 3a. The difference between them is that Figure a shows the solely conducted R1, while Figure b shows the coupled R1R2 reactions. The experiment conducted here is to investigate the influence of R1 and R2 reactions on the encapsulation process. Specifically, Figure a shows the reaction of R1, and Figure b shows the coupled reactions of R1R2. The results show that at low concentration of IP, the limited number of initiation sites (IP) would not help to fabricate DNA cocoons at cell surface. We apologize for the unclear presentation of Figure 3. And for better interpretation of the data, we have improved the explanations in the figure legend, and the detailed discussion has been added in the text.

Figure 3, the background signals are obvious different in Figure 3a, b and c. Besides, please use the same bar length.

Many thanks again for the helpful comment and suggestion. We are sorry for the inconsistent background signals. The change of background is caused by the different assay conditions, where attached and suspended cells are both used in the experiments. Figure a and b respectively show R1 and R1R2 at low IP concentration by using attached cells, while Figure c show the coupled R1R2 reactions at high IP concentration by using suspended cells. So, in the revised manuscript, Figure 3c has been replaced with the figure to show the result by using attached cells. The background signal can thus be unified. And, as suggested, the same bar length has been used in these images.

Figure 1c, it seems that some green signals come from the cell plasma and nucleus region. How to explain this? Were some DNA strands internalized into the cells?

Many thanks again for the comment. Since the negatively charged phosphate backbones of oligonucleotides are prevented from being taken up by cells, the green signals are mainly located at the membrane surface. The contamination could be caused by the integrity damage of cell membrane during nuclei stain prior to encapsulation. To verify the location of these synthesized DNA strands, we used nuclease to digest DNA strands. In this case, signal cannot be obviously detected. In addition, to further confirm the DNA cocoons are fabricated at cell surface, we have constructed 3D visualizations of the encapsulated cells. The 3D vision shows that the fabricated DNA polymer cocoon is surface located. The results have been provided in the updated Figure 1c-iii, which has more clearly shown the encapsulation of cells with DNA cocoons.

Results:

In the experiment, we used nuclease to digest DNA after encapsulation. In this case, the cells were not fluorescent observed. The same result was observed in Supplementary Fig. 11. These observations indicate the synthesized DNA strands are mainly located at cell surface. In addition, for construction a 3D visualization,

confocal scanning of the encapsulated cells were performed at a continue z-axis series. A series of 2-D slice images were acquired and stacked together. The presented 3D images show the DNA cocoons are located at cell surface.

Additional result showing the reconstructed 3D-image of the encapsulated MCF-7 cells.

In Figure 1c (i), I can see many bacteria (cell nuclei were stained by blue color) that are not surrounded by green color. Are these bacteria not encapsulated? If so, it would be misleading to claim the encapsulation efficiency is 95% in Table 1, because perhaps only the encapsulation efficiency of mammalian cells is over 95% and the efficiency of bacteria and yeasts are not discussed in the current manuscript. Additionally, I feel a little confused about the “Biocompatibility” in Table 1. The author claimed that the DNA cocoon is biocompatible, but mammalian cells used in this work lost viability quickly after encapsulation. So what is the meaning of biocompatibility here? And how to quantify it?

We thank the reviewer for the instructive comments and concerns on the two important points, they are very helpful to improve this manuscript.

Firstly, we agree with the reviewer that it is inappropriate to claim 95% encapsulation efficiency in Table 1 as the data are mainly based on the encapsulation efficiency of mammal cells. So we have added the discussion of coating efficiency for bacteria and yeast cells in the revised manuscript. In Figure 1c, as the reviewer notes, faint green fluorescence signals could be co-localized and observed around some of the blue color dots, which indicates these bacteria are encapsulated. But some of the green signals are not well presented in one image since a fixed z-axial plane of confocal scanning was performed. We could see almost all the bacterial have been stained with the green fluorescence during the observation. And some of the blue dots were background signals, the same as that in Figure 1c-ii. In addition, we have given the 3D-reconstructed images of the cells (bacteria, yeast and mammal cells) for a clearer observation of encapsulated cells in DNA cocoons to show high encapsulation efficiency. We thank the reviewer for the help.

Results:

Additional results showing the reconstructed 3D-images of the encapsulated cells.

Secondly, we apologize for the confusion about the statement of biocompatibility in Table 1. Indeed, another reviewer has also raised this important point. Here, the meaning of biocompatibility is that DNA polymerization is a biosynthetic reaction that has been introduced to provide a biosynthetic approach for cell encapsulation, vs. chemical or physical approaches. Some supportive data have been provided in Figure 5 and Table 1. And, after encapsulation, the biocompatibility can be indicated by cell viability data as a function of time (Supplementary Fig. 7 and 12).

Results:

Flow cytometer evaluation showed high viability (95.6%) of the encapsulated MCF-7 cells. After encapsulation, however, viability loss for mammal cell was obvious in a relative short time, with a medium lethal time (LT_{50}) of 11 hours. We tested several possible reasons, including the anchorage-dependent nature of MCF-7 cells, the DNA encapsulation and some reaction conditions. In the experiment, surface-attached and suspended cells without encapsulation were used as controls. We found that LT_{50} for encapsulated cells (both for anchored and suspended cells) were less than 15 hours, and the anchorage-dependent nature of cell was not the major cause of viability loss. Here, it was found that pH value of the reaction buffer was reduced after cell encapsulation. This was observed when we used the cell culture medium containing a pH indicator phenol red. In this case, we tested if the cell viability loss was caused by pH condition changes during the encapsulation process, by using 25 mM HEPES as a buffer agent to maintain physiological pH. By changing assay conditions, we observed the LT_{50} for the suspended and encapsulated cells had been pushed to over 96 hours, indicating the encapsulated cell was kept well after encapsulation. Therefore, the additional assays regarding reaction conditions indicate that the pH change of reaction buffer during the encapsulation process has caused the viability loss. The surface-attached manner of mammal cell and the DNA encapsulation may cause cell viability loss, but not significant within the assay time, as shown in Supplementary Fig. 9. So, the reaction conditions have been optimized during the encapsulation process. The details have been specified in the Methods section and Supplementary Table 2, and corresponding discussion has been added to the text.

Results:

Additional results in Supplementary Fig. 9 showing the viability tests of the encapsulated MCF-7 cells as a function of time, showing the influence of different conditions on cell viabilities. The anchored and suspended cells without encapsulation are used as controls.

Overall, I cannot recommend this paper to be published in Nature Communications.

In summary, we have made a substantial improvements of this work based on the reviewer's helpful comments and suggestions, and also addressed the critical points raised by other reviewers. We think these improvements would give a clearer and more significant presentations to meet the journal requirements, and this work would be of interest in a broad range of biological research fields. And again, we thank the reviewer for the great help.

Reviewer #3 (Remarks to the Author):

In this manuscript, the authors developed the in situ DNA-oriented polymerization approach to fabricate DNA polymer networks onto three kinds of cells. And they applied this method to specifically capture and release encoded cells in a DNA patterned slide surface. This work is novel and publishable, but the manuscript could be improved.

Many thanks for the positive comments.

1. The schematic illustrations of the principles of isDOP showed that the LatDNAs connect different LonDNAs together. However, the AFM images in Fig.2 did not prove the connection between LonDNAs. And the agarose gel analysis actually confirmed the repetitive double-stranded DNA structure, but it did not show the

crosslink between different LonDNAs. Please give more supports and explanations about the network formation.

Many thanks for the helpful comment. We agree with the reviewer that the cross-linkage of LatDNAs and LonDNAs lays the foundation of DNA cocoon fabrication. As indicated by the reviewer, although the AFM observations and the fluorescent observations give supportive information of the fabricated DNA network, the detailed information are insufficient, especially for intercrossing linkage among different LonDNAs. We have also realized that these experiments are mainly conducted in the solution or at the surface of mica, instead of on cells. So, to address these points, additional and new experiments have been conducted at cell surface to investigate the details. And more results have been added to Fig. 2 and Supplementary Results.

Firstly, we have fabricated DNA strands on cells and digested them with S1 nuclease, and brought the products to gel analysis. A DNA ladder tails were observed (Supplementary Fig. 6), indicating that the repetitive double-stranded DNA structures of different length were generated. The length polymorphism of these DNA structures in the gel ladder tails is an important clue that the two DNA strands are crosslinked. Otherwise, a relative single band with large molecular weight would be observed after S1 digestion. If the LonDNAs are not bridged by LatDNAs, long double-stranded structures would be synthesized by R2 reaction. In addition, the product of R1R2 has shown extremely large molecular weight that they were stuck in the loading well, indicating the two strands have been cross-linked. Therefore, the additional experiments on cell have further confirmed the network formation. The result has been provided as Supplementary Fig. 6. And more discussions have been added to the text.

Secondly, a more specific experiment has been conducted to reveal the relationship of the two strands. As indicated in the schematic illustration, the two primers (IP and BP) respectively initiate the two reactions (R1 and R2) to synthesize the two strands (LonDNA and LatDNA). So, we have respectively labeled the two primers with fluorescent dyes (green and red) as the indicators to monitor the locations of LonDNA and LatDNA. After encapsulation, the synthesized DNA strands were stained by the dye-labeled primers. We found the two fluorescent signals were almost co-localized, shown below. Therefore, these DNA strands have been interacted to form DNA cocoon at cell surface. The additional results have been added to the main text as Fig. 2e, and additional discussions have been provided in the main text. We thank the reviewer for the great help.

Additional results in Fig. 2e. The scanning confocal microscope images showing the locations of LonDNA and LatDNA strands at cell surface. The two strands were respectively labeled with dye-modified IP (green) and BP (red). Co-localization of the two fluorescent signals reveals cross-connections of the LonDNA and LatDNA strands. Scale bars, 10 μ M.

2. From the fluorescent scanning images, the DNA-embed cells formed both aggregates and single one, and the cell aggregation could affect the precise manipulation. Please give the results about the single cell encapsulation ratio.

Many thanks again for the helpful comment and suggestion. We agree with the reviewer that cell aggregates could affect the precise manipulation of the encapsulated cells. So we calculate single cell encapsulation ratio based on the flow cytometry evaluation, which is that 87.18% of the mammal cells are individually encapsulated, as shown below.

Additional scheme and result in Supplementary Fig. 8 showing the principle of the method for the evaluation of single cell encapsulation by cytometry analysis, and the corresponding result by FSC-A vs. FSC-H plotting analysis.

Methods and Results:

According to the reported method (Nature Immunology, 2009, 385-393), single cell and cell clumps could be discriminated by recording the FSC-A and FSC-H parameters. This strategy is based on the assuming that the instrument is set that A/H is constant, when correcting the same parameter (for instance, FSC) A vs. H, the single cell will have the constant A/H value. Therefore, the encapsulation ratio of

single cell could be evaluated by FSC-A vs. FSC-H plotting. The result showed an averagely 87.18% of the detected events were single encapsulated. The result has been provided as Supplementary Fig. 8.

3. As mentioned in cell vitality test part, the vitality of DNA cocoon embed mammalian cells could not keep well. In this case, the biocompatibility of this approach for mammalian cells needs to be reconsidered.

Many thanks again for the helpful suggestion. We agree that the vitality of DNA cocoon embed mammalian cells is not keep well after encapsulation. As the reviewer notes, the encapsulated cells lost viability in a relative short time (<20 h) as compared to that of the other two cells (several weeks or longer). So, we have performed additional experiments to find out the potential causes of the viability loss. After eliminating possible causes, we have found that the major cause of this phenomena is the pH decrease during DNA polymerization. So the reaction buffer has been added with a buffer agent to maintain the physiological pH during the encapsulation process. As a result, this encapsulation approach has been improved with optimized conditions. The results have been provided in Supplementary Fig. 7 and 12 in the revised manuscript. So the assay conditions has been specified in Methods section and Supplementary Table 2, and corresponding discussion has been added to the text. We sincerely thank the reviewer for the help.

Results:

We first tested if the unexpected decrease of mammal cell viability was due to the anchorage-dependent nature of animal cell, or the DNA encapsulation would cause cell viability loss. We introduced adherent cells and the cells without encapsulation as the controls, respectively. The results showed the influence of the anchorage-dependent nature of mammal cell was limited. But the DNA encapsulation caused significant viability loss. Previous reports have shown that the DNA strands or their assembled-structures are relative biocompatibility when they were attached to cell surface (Nat. Nanotechnol. 2017, 12, 453-459; Nat. Methods 2015, 12, 975-981; Nat. Nanotechnol. 2017, 12, 1183-1189), so there may be other reasons. During the experiment, we also found pH value of the buffer was significantly reduced when we used the DMEM containing a pH indicator (Gibco 11960044) in the additional experiment. This was not previously observed as we used DMEM without the pH indicator before. To test if the cell viability loss was significantly affected by pH value change, we used a buffer agent (25 mM HEPES) to maintain physiological pH as the control. In this case, the medium lethal time (LT₅₀) of the encapsulated could be increased over three days. The results indicated the pH value change during the encapsulation process was the major cause.

4. Some previous works about DNA materials for cell encapsulation are not mentioned in this manuscript. Here are some examples.

- 1. *Nature Methods* 2015, 12, 975-981.**
- 2. *Advanced Materials* 2013, 25, 4714-4717**

Many thanks for the suggestion, so we have added related references in the text to enrich the research background, please see Ref. 26, 33 and 49-51. So the order of these references has been updated. And again, we thank the reviewer for the great help.

REVIEWERS' COMMENTS:

Reviewer #1 (Remarks to the Author):

The authors have added adequate data to address my comments about cell viability.

Reviewer #3 (Remarks to the Author):

I am satisfied with the revision.

REVIEWERS' COMMENTS:

Reviewer #1 (Remarks to the Author):

The authors have added adequate data to address my comments about cell viability.

Many thanks for the help.

Reviewer #3 (Remarks to the Author):

I am satisfied with the revision.

Many thanks for the help.